# Experiences along the diagnostic pathway for patients with advanced lung cancer in the USA: a qualitative study

Morhaf Al Achkar [1], Monica Zigman Suchsland [1], Fiona M Walter [2], Richard D Neal [3], Bernardo H L Goulart,[4] Matthew J Thompson[1]

[1]Department of Family Medicine, University of Washington, Seattle, Washington, USA
[2]Department of Public Health and Primary Care, University of Cambridge, Cambridge, UK
[3]University of Leeds, Leeds, UK
[4]Food and Drug Adminstration, Washington, DC, USA

**Correspondence to**
Dr Morhaf Al Achkar;
alachkar@uw.edu

## ABSTRACT

**Background** Most patients with lung cancer are diagnosed at advanced stages. However, the advent of oral targeted therapies has improved the prognosis of many patients with lung cancer.

**Purpose** We aimed to understand the diagnostic experiences of patients with advanced lung cancer with oncogenic mutations.

**Methods** Qualitative interviews were conducted with patients with advanced or metastatic non-small cell lung cancer with oncogenic alterations. Patients were recruited from online support groups within the USA. Interviews were conducted remotely or in person. Analysis used an iterative inductive and deductive process. Themes were mapped to the Model for Pathways to Treatment.

**Results** 40 patients (12 male and 28 female) with a median age of 48 were included. We identified nine distinct themes. During the 'patient interval', individuals became concerned about symptoms, but often attributed them to other causes. Prolonged or more severe symptoms prompted care-seeking. During the 'primary care interval', doctors initially treated for illnesses other than cancer. Discovery of an imaging abnormality was a turning point in diagnostic pathways. Occasionally, severity of symptoms prompted patients to seek emergency care. During the 'secondary care interval', obtaining tissue samples was pivotal in confirming diagnosis. Delays in accessing oncology care sometimes led to patient distress. Obtaining genetic testing was crucial in directing patients to receive targeted treatments.

**Conclusions** Patients experienced multiple different routes to their diagnosis. Some patients perceived delays, inefficiencies and lack of coordination, which could be distressing. Shifting the stage of diagnosis of lung cancer to optimise the impact of targeted therapies will require concerted efforts in early detection.

## BACKGROUND

Lung cancer is the leading cause of cancer death and the second most common cancer type in the USA.[1] In 2016, the incidence of new lung cancer cases in the USA was 56 per 100 000 people and the rate of lung cancer death surpassed the rate of any other cancer death, with 38.5 per 100 000 people.[2]

### Strengths and limitations of this study

⇒ The study's strengths include exploring the perspectives on the diagnosis journey of a large number of participants representing a relatively new group of lung cancer survivors: those on targeted therapies who experience significantly superior outcomes.
⇒ Our findings were developed within an existing theoretical framework used in research on early cancer diagnosis by many other countries.
⇒ The study's limitations include relying on individuals identified from lung cancer survivor groups, which may have reduced the representativeness, particularly of individuals from less affluent backgrounds.
⇒ Only a small proportion of our participants experienced barriers to accessing care due to financial concerns, which may have limited our ability to determine these factors' impact.
⇒ Recall bias and differential recall bias are major concerns with this type of research.

Although screening for lung cancer using low-dose CT scanning has been recommended in the USA since 2013, the majority of individuals are diagnosed either after seeking clinical care with symptoms or as an incidental finding after imaging.[3] The poor outcomes associated with lung cancer are at least partly the result of the length of time between a patient first experiencing bodily changes and being diagnosed.[4–7] Based on a pooled analysis of 56 studies, the median time from symptom onset to diagnosis ranged from 41 to 143 days.[8] Unfortunately, a significant proportion of individuals with lung cancer are at advanced stages at the time of diagnosis and have an overall survival rate measured in months.[9]

There has been surprisingly little US research on patients' perceptions of the diagnostic pathways for lung cancer. Most research assessing time to diagnosis has been performed in European healthcare

systems and in smokers, making comparisons with the US population or with non-smokers difficult.[10 11] There has been almost no research on the diagnostic experiences of patients with advanced lung cancer who are receiving targeted therapies for oncogenic mutations such as c-ros oncogene 1 (*ROS1*) mutations (1%), anaplastic lymphoma kinase (*ALK*) rearrangements (3%–7%) and epidermal growth factor receptor (*EGFR*) mutations (10%–15%).[12] Targeted therapy has improved the outcomes of patients with these mutations, with median overall survival times of 52.1 months for *ROS1*, 81 months for *ALK* and 29.7 months for *EGFR*. Thus, understanding the pathway to diagnosis is especially important in this population.[13–16]

The purpose of this study was to explore the experience of the diagnostic process among patients with advanced lung cancer whose tumours tested positive for oncogenic driver mutations in order to identify potential areas to improve the efficiency and experience of the diagnostic pathway.

## METHODS
### Study design
This qualitative study used indepth individual patient interviews.

### Study population
Participants met the following inclusion criteria: (1) histological or cytologically confirmed diagnosis of metastatic or advanced non-small cell lung cancer (NSCLC) with the presence of one oncogenic alteration (*EGFR*, *ALK* or *ROS1*); (2) physically and psychologically well enough to participate; (3) proficient in English; and (4) receiving care in the USA. We identified patients using online oncogene-focused lung cancer support groups. Detailed methods are included in a previous publication.[17]

### Study procedures
Participants were interviewed by phone, video conference or in person depending on location and preference. One author (MAA) conducted the interviews after receiving verbal consent. Interviews were audio-recorded and transcribed verbatim. Participants were asked to describe their diagnostic journey from the moment of first noticing symptoms to initial treatment. The interviewer asked follow-up questions for clarification. Participants were given a $50 gift card for participating. Interview questions and follow-up prompts are included in online supplemental appendix 1.

### Analysis
NVivo V.11 was used to organise the data and conduct the analysis. Inductive and deductive thematic analysis was applied. As outlined by Carspecken,[18] the transcripts were read by the lead author (MAA) and low-level codes were developed. The codes were then collated by topic. Codes were mapped following the Model for Pathways to Treatment (figure 1).[8 19 20] Themes and subthemes emerged through an iterative process, and all authors engaged in peer debriefings as groups and

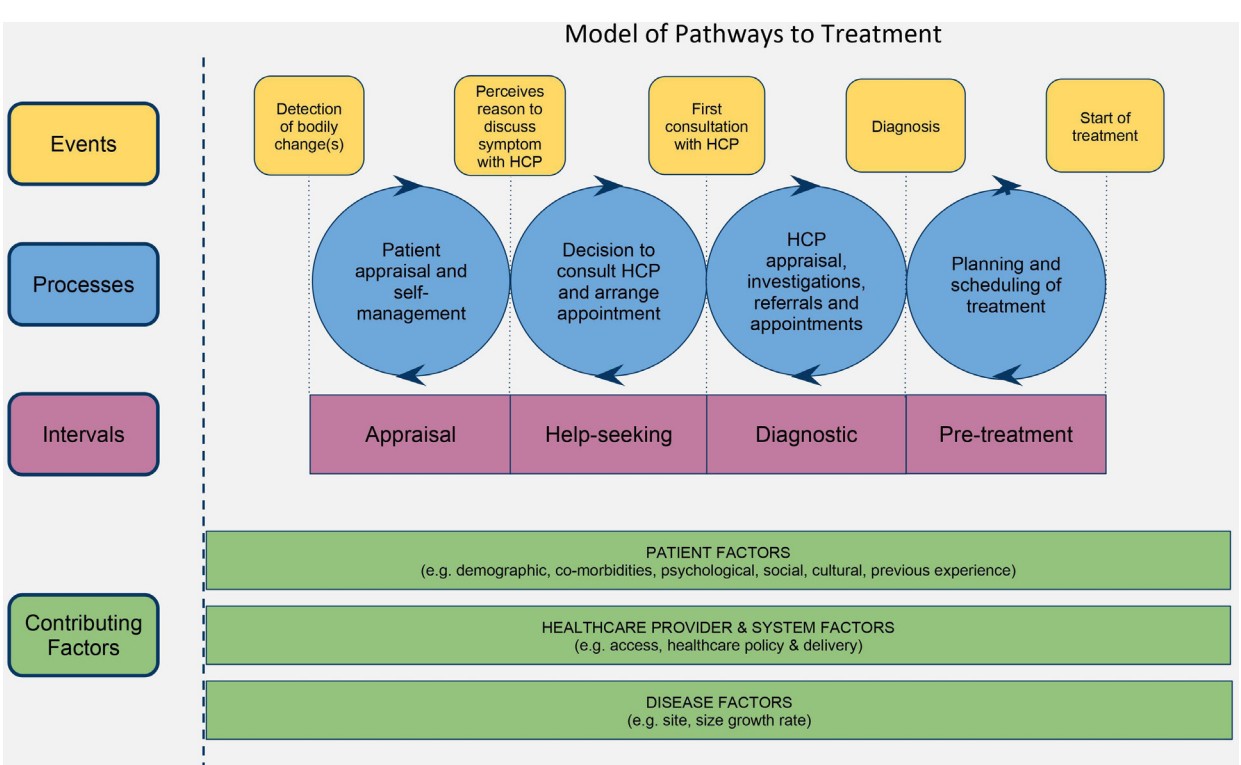

**Figure 1** The conceptual Model for Pathway to Treatment. HCP, healthcare provider. Reproduced with permission of SAGE Publications Ltd., London, Los Angles, New Delhi, Singapore and washington DC, from Walter FM *et al*.[19]

dyads reviewing aspects of the work, including coding and analysis, theme development, and description of findings. Themes were organised based on the *Aarhus statement on cancer diagnostic research* stages: patient interval, primary care interval and secondary care interval.[21 22] Transcripts and themes were reviewed and synthesised to characterise the different types of diagnostic pathways experienced by patients.

MAA is a patient with stage IV, *ALK*-positive lung cancer, a family doctor and a qualitative researcher. MZS is a researcher with experience in qualitative research. MT is a family physician in the USA with extensive research experience on disease diagnosis. BHG is an oncologist and health service researcher. FMW and RDN are primary care lung cancer researchers from the UK. MAA performed the main analysis and engaged in peer debriefing with coauthors as dyads and groups. The coauthors reviewed aspects of the work, such as analysis and coding, theme development, and writing the results.

### Patient and public involvement
The main author is a patient with stage IV lung cancer and a member of one lung cancer support group. The research questions were informed by conversations with lung cancer communities. Patient gatekeepers helped in recruiting participants by sharing about the study in their support groups. The study will be shared with cancer communities on social media and specifically in support group venues.

## RESULTS
A total of 40 patients were interviewed. Their mean age was 48 (range 30–75); 12 were male and 28 were female. Interviews were conducted for a median of 19.5 months (range 3–152) after diagnosis (table 1). All participants had a primary diagnosis of metastatic or advanced NSCLC with one driver oncogenic alteration. We noted seven different diagnostic pathways experienced by patients, rather than a single course. These pathways varied primarily by the initial presentation site (primary care, emergency room (ER) and so on) due to the perceived urgency of symptoms (figure 2).

### The experience of lung cancer diagnosis
Emergent themes within the diagnostic intervals (patient, primary care and secondary care) are detailed in the following sections.

#### Patient interval
##### Initial concerns about symptoms despite low perception of risk
Prior to diagnosis, lung cancer did not come to mind for most participants, especially as most were younger and non-smokers. Many believed their healthy lifestyle protected them against such illnesses. In contrast, those who smoked suspected lung cancer from the onset of symptoms. The participants recalled experiencing various

**Table 1** Participant characteristics

| Participant characteristics | Median (range)/ count |
|---|---|
| Age | 49 (30–75) years |
| Gender | |
| Male | 12 |
| Female | 28 |
| Race | |
| White | 34 |
| Others (Asian, Hispanic, biracial (Asian and Hispanic)) | 6 |
| Region in the USA | |
| West | 18 |
| Northeast | 8 |
| Midwest | 7 |
| South | 6 |
| Insurance | |
| Private | 34 |
| Medicare | 4 |
| Medicaid | 2 |
| Time since diagnosis | 19.5 (3–152) months |
| Cancer stage at time of interview | |
| IV | 38 |
| IIIb | 2 |
| Mutation | |
| ALK | 20 |
| EGFR | 14 |
| ROS1 | 6 |

*ALK*, anaplastic lymphoma kinase; *EGFR*, epidermal growth factor receptor; *ROS1*, c-ros oncogene 1.

new symptoms or a change in persisting symptoms that concerned them. Most reported non-specific symptoms; some were respiratory in nature, while others related to organs and systems due to metastatic spread (eg, bone pain) or were constitutional (eg, fatigue, weight loss). Some recalled the symptoms being present up to a few months prior to diagnosis. A minority did not recall any symptoms. Diagnosis occurred after imaging for other reasons, such as an injury or trauma (box 1).

##### Attribution of symptoms to other causes and not always seeking care immediately
Participants initially attributed their symptoms to reasons other than lung cancer. Coughing, for example, was explained by forest fire smoke in the air; back pain was attributed to muscle spasm; fatigue was blamed on depression; and shortness of breath with activities on excessive weight. Even haemoptysis raised concern for tuberculosis as a more likely cause. Many participants did not worry initially because the symptoms were perceived as mild or they felt others had similar symptoms, such as

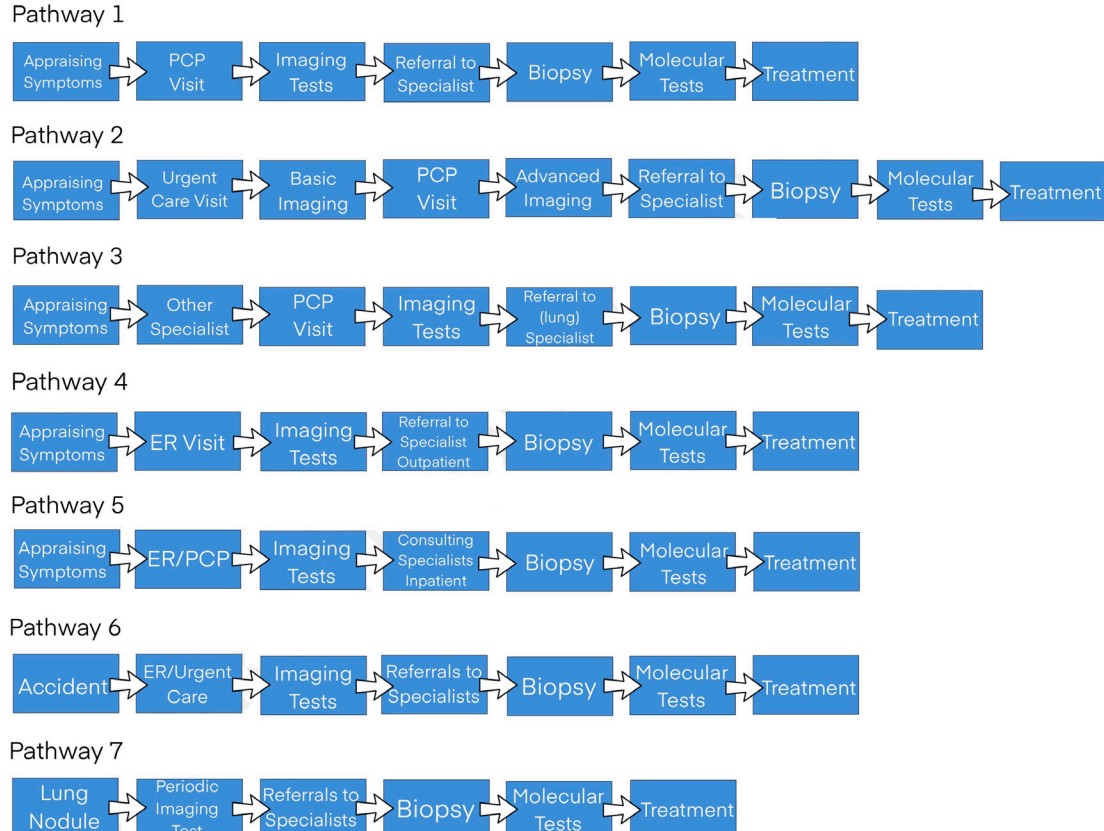

**Figure 2** Identified pathways to diagnosis. ER, emergency room; PCP, primary care provider.

dismissing a cough during influenza season. Finally, some people did not have health insurance at the time of early symptoms and the potential cost of healthcare services deterred them from seeking help.

### Changes in severity or nature of symptoms prompting care-seeking actions

Participants expressed experiencing a change in their level of concern prompting them to seek medical attention. Reasons included symptoms getting worse, especially after initially improving; not responding to treatments for other suspected illnesses; symptoms lingering; disruptive pain; symptoms developing in combination; alarming symptoms appearing, such as haemoptysis or significant weight loss; and symptoms affecting quality of life or affecting sleep. Sometimes family members or friends had advised the person to seek care after noticing symptoms.

Most individuals initially visited their primary care providers (PCPs) to get help with their symptoms or to determine the reason for the symptoms that had become concerning. Some first visited urgent care, especially when they encountered delays in accessing a PCP. Some patients who had established relationships with specialists consulted with them first; some complained to their ear, nose and throat doctor about their haemoptysis, while others complained to their gastroenterologist about their shortness of breath.

### Primary care interval
#### Doctors initially treated for illnesses other than lung cancer

Participants described that providers were not alarmed by, or sometimes dismissed, their initial symptoms. For many, the initial course of management was the investigation and treatment of benign aetiologies. In some cases, initial investigations supported other diagnoses, such as a respiratory infection from chest X-ray (CXR) or acid reflux confirmed on endoscopy. In other cases, initial tests were normal. Some patients' symptoms were attributed to and treated as other diseases, for example, a shortness of breath was attributed to underlying asthma and treated with inhalers and steroids. Some patients were referred to specialists, such as physical therapy or orthopaedics for musculoskeletal complaints. The wait for specialist appointments sometimes took several weeks. Not infrequently, providers used 'safety netting', or contingency plans, such as scheduling return visits, follow-up CXR and trying other treatment plans (box 2).

#### Discovery of imaging abnormality, often on CXR and/or chest CT, leading to diagnosis

A major turning point identified by some participants was getting a CXR, either at their request or prompted by their PCP, intended to identify the cause of symptoms. Imaging studies were also ordered when treatment failed or to assess whether previously noted radiological findings had been resolved. Occasionally, imaging tests

## Box 1    Supportive quotes for patient interval

**Initial concerns about symptoms despite low perception of risk.**

⇒ "I have not been into a doctor for a medical check-up at all in all that time. I never had any days off taken my entire work experience." (1001)

⇒ "I looked really healthy and I'm not a smoker." (3005)

⇒ "I started seeing symptoms three-four months before diagnosis. I noticed some tightness in my chest." (1003)

⇒ "I just had a dry cough that would not go away." (2007)

**Attribution of symptoms to other causes and not always seeking care immediately.**

⇒ "There were a lot of forest fires. The air was always really smoky and I thought maybe part of the headaches or not feeling quite right was caused by the smoke." (2013)

⇒ "I was having some lower back pain in the kidney area and had some other symptoms that made me think maybe I have got kidney stones." (2006)

⇒ "Everybody else in the family also seemed to have flu-like illness going on with a cough; cold-cough kind of thing." (1005)

⇒ "I was very weak, very lethargic; the worst I ever felt in my life. I tried to self-medicate. I was not insured." (1003)

**Changes in severity or nature of symptoms prompting care-seeking actions.**

⇒ "Three more weeks went by and the cough continued to get worse to the point where my chest started hurting and I had a little bit of a backache." (1005)

⇒ "My wife came back from China, she was away for about a month. She said, "Your coughing is different." At the time, I didn't notice anything yet." (2012)

⇒ "I coughed a little blood. I am not stupid I knew I had big trouble. There was no question; I called the doctor." (1012)

⇒ "I decided, I'm going to go ahead and see my primary care physician to see if maybe she had some more suggestions of what I can do to help this throat situation." (1017)

## Box 2    Supportive quotes for primary care provider interval

**Doctors initially treated for illnesses other than lung cancer.**

⇒ "I recall going to see the Primary Care Physician and mentioned, "I'm constantly clearing my throat." They casually dismissed me; the symptom continued." (3002)

⇒ "I went to the doctor and she did full blood work and said everything looks great. She said the cough was probably just a little bit of a remnant from the cold and typically it can take 3, 4, 5 or even 6 months to go away and not to worry about it too much." (1001)

⇒ "I went back to my doctor again and said, "okay, we've tried asthma, we've tried the allergy, here is some reflux medications," which kind of helped. She sent me to my doctor that specializes in reflux. We did an endoscopy. They came back with, "you do have reflux."" (3004)

⇒ "I kept seeing various doctors and they would always send me home. Like, "Oh, it's a seasonal cold. Oh, it's allergies. Oh, you pulled the muscles from coughing too much, here are some steroids."" (1008)

⇒ "I went to a walk in clinic two different times and was diagnosed with walking pneumonia. Both of those times, I did have an x-ray of the chest, and it just showed some cloudiness, it didn't show any kind mass." (2007)

⇒ "She put me on a different prescription but she said, "If you're not better in a couple of weeks, come back and we'll do a full pulmonary workup and we'll do more diagnostic testing 'cause this was concerning."" (3001)

**Discovery of imaging abnormality, often on CXR or a chest CT, led to diagnosis.**

⇒ "The doctor gave me steroids was leaving the room, I said something to the effect of, "I thought I would have to get an X-ray." I'm the one who mentioned the word, "X-ray."" (1017)

⇒ "I went to get an x-ray of my left rib cage. It felt like something was there. I told my doctor that I think I have cancer and I want her to check for cancer. So she obliged." (1009)

⇒ "I made an appointment and set me for a chest x-ray. And this is was to me really an important point. There was a radiologist sitting in the booth. He looked at me and from the look on his face I just knew." (1011)

⇒ "After the car accident I was taken to a trauma center and they scanned me and said, "You have a broken back and lung cancer."" (2009)

⇒ "I went back to the doctor the next day and she took a look and she said, "Hmm, I don't like that (swelling in supra-clavicular area)." And she sent me for an ultrasound." (3001)

⇒ "I went for a physical to my primary care doctor. He noticed that I had motor deficits in my hands. He suggested that I get an MRI. I actually had to go and see a neurologist in order to get the prescription for an MRI and paid for." (2013)

⇒ "As soon as the order went in for the chest x-ray, I went in to have it done. That night my doctor called back and said, "we saw some things on the chest x-ray, we want to get you in for CT scan." So the chest x-ray was a Monday, the CT scan was a Thursday. On the night of the CT scan, she called back and said, "It looks like cancer."" (3004)

⇒ "She noticed that my breath sounds weren't right. So she ordered a CT and called me the next day and told me that she was going to send me for a PET. She was pretty concerned that it was lung cancer." (1004)

Continued

were used to evaluate incidental conditions such as injuries, while other patients received CXR to follow up on nodules seen on previous imaging. Other imaging tests used to evaluate symptoms elsewhere in the body identified lung cancer as an incidental or unexpected finding, such as MRI for back pain or breast-screening MRIs identifying lung lesions.

For many patients, a diagnosis of lung cancer was supported by a chest CT done after an abnormal CXR or to discover the primary site after a metastasis was found. Scheduling the CT scan was often rushed. Sometimes PCPs pushed for this to happen or, when scheduling was delayed, advised patients to go to the ER.

*Severity of symptoms prompting need for emergency care*

Some patients went directly to the ER with distressing symptoms such as severe shortness of breath. Others sought care in the ER for symptoms such as headache and back pain as they had no PCP. At times, the patient's condition deteriorated quickly, requiring admission due to hypoxia or losing consciousness with brain tumours causing seizures. Occasionally, delays in diagnostics or the

**Box 2  Continued**

**Severity of symptoms prompting need for emergency care.**

⇒ "I was scheduled for a CT scan but the next opening wasn't for like 2 or 3 weeks. I was having so much coughing that I couldn't speak or breathe properly. So I called my healthcare provider's office. She advised that I should go to the ER and get a CT scan." (2007)

⇒ "We scheduled the biopsy for Thursday. Tuesday morning before I could go for the biopsy, I woke up coughing up blood, a considerable amount of blood which was new that it never happened. So I drove myself to the ER." (2008)

⇒ "The second I went in the pulmonologist office, he checked my oxygen and it was 85%. I took his advise and went to the hospital." (1014)

CXR, chest X-ray; ER, emergency room; PET, positron emission tomography.

perception that their PCP could not offer much besides office testing prompted the patient to go to the ER. Other patients were advised to go to the ER after findings such as a pulmonary embolism or massive brain metastasis. At the ER, it was not uncommon for the patient to be admitted. Some patients demanded urgent consultations from specialists and to be admitted to complete the cancer work-up and start treatment.

### Specialty care interval
#### *The pivotal nature of tissue sample collection*

Once imaging raised the alarm for cancer, interventional radiologists, pulmonologists or thoracic surgeons obtained tissue samples. While some patients saw a specialist fairly quickly, others experienced significant delays. Bronchoscopy, needle biopsies, sampling of pleural effusions and occasionally surgical biopsies were used to clarify if the lesions seen on imaging were cancer, to identify the type of cancer and to obtain tumour tissue for genetic testing. Results were delivered within a few days. While a bronchoscopy was often uneventful, it sometimes led to major bleeding, collapsed lungs or the patient requiring resuscitation. Occasionally, concerns over the procedure led to delays in this diagnostic step. When decisions were made to forego biopsy, patients felt they were provided false reassurance based on less reliable information, such as the appearance on images and their overall assumed low risk for cancer (box 3).

#### *Access to oncologists determined staging but perceived delays led to distress*

Patients were referred to an oncologist once diagnosed. The referral was made urgently, often by the PCP or pulmonologist based on imaging findings or following pathology results. It was not uncommon for patients to perceive a delay in making appointments, causing frustration. To identify the right specialist and overcome delays, patients often leveraged personal connections or sought help from family and the cancer community. First meetings with oncologists often involved reviewing the results and setting treatment plans. These were usually short, especially if molecular results were not back. Oncologists

**Box 3  Supportive quotes for secondary care interval**

**The pivotal nature of tissue sample collection.**

⇒ "The PCP said, "I think you have a problem. You got to go and see a Pulmonologist immediately." Finding a Pulmonologist with an opening is impossible." (3002)

⇒ "She said it looks like a metastatic disease. She set me up with a biopsy of the lung and a biopsy of the liver." (3003)

⇒ "I tried to have a lung biopsy done and I was sitting on the table and the radiologist came in and he said, "I can't biopsy that nodule, no way." The team were all arguing about it over me and finally the radiologist said it is not biopsiable and so I left. They said, well, that probably is not cancer." (1011)

⇒ "I had a biopsy of the lungs and ended up with a completely collapsed lung and a chest tube." (1006)

⇒ "A senior pulmonologist said, "We suggest drain her lung, drain her effusion and let her out." But the hospitalist was like, "I don't want to let her out until we get a biopsy because she's going to be in the community and it's trying to schedule all of these and she's going to be given and run around and this is an emergent case so I'm leaving her until she can get the biopsy."" (1019)

⇒ "I had a needle biopsy and he called me a couple of days later, "It's lung cancer and it's adenocarcinoma, and I'm going to send you to an oncologist." (3001)

**Access to oncologists determined staging but perceived delays led to distress.**

⇒ "I was discharged from the hospital, came home, had a follow-up appointment a few days later with an oncologist who was just part of the healthcare system. They just assigned to me to somebody." (2008)

⇒ "I was leaving ever more frantic messages and calling again and again and pushing the reception desk. It was about quarter to 12 before I finally convinced her I needed to talk with the doctor today rather than wait, find out how long I might live." (2014)

⇒ "I was able to find a lung cancer foundation. And one of the folks there told me about a lung cancer oncologist in an area close to me and said, "You should reach out to them. And tell them I told you to give them a call." And so I did just that. And the doctor called me back." (3002)

⇒ "They noted that there were tumors spotted on in my neck region and at that point in time, they wanted to do a full PET scan to figure out what the extent it was. They turned around really quickly. I must say after the original scan, the quickness of my treatment and exploratory work was very fast." (1013)

⇒ "I had developed what I had thought was sciatica, but when they did the scan they found out that it was metastasis in my bone that was hitting my sacrum that was kind of causing the sciatic nerve to be inflamed." (2001)

**Genetic testing was crucial in directing patients to targeted treatments.**

⇒ "I'm grateful my oncologist ordered molecular testing. I know that's becoming standard as care now, it was not quite so much standard as care at that stage." (1001)

⇒ "So the following week we are supposed to have an appointment but the insurance took a while to approve everything. We postponed that appointment till we got result and the result were *ALK* positive." (1019)

⇒ "When week number 4 went around, (the local oncologist) has not been following up with me. I've been calling you, we still don't have results. I was uncomfortable right around week 6, so I flew and sat

Continued

## Box 3   Continued

down with an oncologist (at a major cancer center) and he basically said well we don't need to wait. Let's do blood test (liquid biopsy). I'll have the results for you in 7 days." (1020)

⇒ "I think that somebody dropped the ball at the hospital because the request for the testing wasn't sent until three weeks after they did the surgery they hadn't even requested to do the molecular testing. So when they finally did it still took another few weeks." (1018)

⇒ "I had a week of radiation and they were still waiting for the mutation to come back." (3006)

⇒ "He wanted me to start chemo treatment immediately because it seemed to be very aggressive whatever this was. Without waiting for the results of any genomic testing and this is still a point of concern for me because, looking back, I feel things work done improperly. We did not wait for the results of the genomic testing. I was started on chemotherapy." (2008)

⇒ "The surgery basically gave me a hug and said, there's some really good news, the tumors tested positive for *ROS1*, and I had no idea what this means." (3002)

⇒ "I'll never forget when my doctor came in and he said, "Hey, you have the *ALK* mutation." And he said, "You're really lucky." And I'm like, "What do you mean? How am I lucky?" And he was like, "We have this great medicine that was just approved by the FDA."" (1008)

ALK, anaplastic lymphoma kinase; FDA, Food and Drug Administration; PCP, primary care provider; PET, positron emission tomography; *ROS1*, c-ros oncogene 1.

often completed the diagnostic work-up by ordering additional imaging such as positron emission tomography (PET) scans or brain MRIs. Since our participants had advanced diseases, PET scans often showed metastasis outside the lungs.

### Genetic testing was crucial in directing patients to targeted treatments

For our participants, molecular testing on tissue or blood samples was obviously an instrumental part of their diagnosis. Realisation of a positive mutation was met with relief, as patients were fortunate to be a candidate for targeted therapy. However, molecular testing results sometimes took several weeks or were overlooked by providers. Looking back, some patients described frustration at being given chemotherapy instead of waiting for molecular testing results. Some, however, needed emergency chemotherapy, radiation or surgery to relieve symptoms.

## DISCUSSION

As the first on the subject, this study contributes to the literature on pathways to diagnosis and the intervals of diagnosis among patients with advanced lung cancer on targeted therapy. The participants were mostly young, non-smokers, unlike those in previous research in this area. We used a well-established model to map participants' experiences from their initial realisation of

symptoms, through contact with healthcare, and diagnostic workup.[19 20]

Previous studies on this 'patient interval' suggested that atypical or vague symptoms caused delays in knowing when to seek care. Previous research (with participants who were predominantly smokers) noted reluctance among patients to visit their healthcare provider when symptoms emerge,[6] but this pattern was not reported by the majority of our study participants. Because they were younger than the average age at presentation of lung cancer and/or presented with non-specific symptoms,[4] their concerns were typically attributed initially to benign diseases. Recognising the symptoms and making a diagnosis can be particularly challenging when a patient has comorbid conditions with symptoms similar to those of lung cancer.[4 23]

Many patients perceived inefficiency and delays in the primary care interval. However, these perceptions were made retrospectively, bringing into question whether an actual delay took place. Some patients felt they had to advocate for themselves to obtain initial diagnostic testing and push for more advanced testing when initial tests were inconclusive. This finding is consistent with the role of self-advocacy in improving the quality of care for patients with cancer.[24 25] Previous studies suggested dismissive responses from PCPs may impact patients' decisions to consult care again.[26 27] In contrast, our participants reported persistence and at times sought other providers. Some providers clearly had contingency and follow-up plans, but patients commonly felt they were dismissed without clear 'safety netting'.[28]

Previous US studies of patients with lung cancer have suggested delays occur mainly in the primary care interval through misdiagnosis (and from monitoring nodules) rather than in the specialty care interval.[29] In contrast, difficulty in accessing secondary care is a major cause for delays in the UK.[6] Our study found that patients' sense of urgency and perception of unnecessary waiting intensified after receiving imaging diagnosing possible cancer. Many complained about delays in accessing pulmonologists, oncologists or in results from molecular testing. While these waits were fairly short and probably had little impact on the overall prognosis, they did appear to intensify patient emotion.

This study has many strengths. It is the first to explore the perspectives of a relatively new group of lung cancer survivors: those on targeted therapies who experience significantly superior outcomes. Interviewees may have been better able to reflect on their diagnostic journey in the absence of side effects from chemotherapy or radiation therapy. Our findings were developed within an existing framework used in research on early diagnosis of cancer by many other countries. Our study also has a few limitations. Only a small proportion of our participants experienced barriers to accessing care due to financial concerns, which may have limited our ability to determine the impact of these factors.[29] Our sampling relied on individuals identified from lung cancer survivor

groups, which may have reduced the representativeness, particularly of individuals from less affluent backgrounds, and over-recruited patients who were more engaged with their disease and diagnostic work-up. Also, we did not actively seek to define smoking status during the interviews, thus we omitted characterising the sample by this factor. Finally, as a qualitative exploration, our study was not equipped to provide insights about frequencies of occurrences, time indicators or variations between participants based on their characteristics.

Our study has important practical implications. First, lung cancer affects everyone, including those thought to be at low risk. The public must be made aware of this so when new symptoms appear they will seek healthcare promptly. This advice should be tempered with knowledge of the extremely low probability of cancer in most patients and the poor predictive value of most symptoms. Second, PCPs should be vigilant for rare but serious diseases with similar symptom profiles to benign conditions. 'Safety netting' should include sharing diagnostic uncertainty and encouraging patients to return for further assessment when symptoms fail to respond. More precise diagnostic tools would be valuable to PCPs in this difficult task, but ready access to CXR and CT is clearly important. Third, while access to secondary care for serious conditions like cancer may not be a challenge for all patients in the USA, the need for coordinating care, communication with patients and provision of up-to-date standards of practice continue to be an issue. This issue is relevant especially to patients with lung cancer where targeted therapy has changed the disease outcomes in the past few years for patients who have received molecular testing. It is paramount that these new standards of care be available promptly to all patients.

**Acknowledgements** The authors thank LUNGevity, especially Upal Basu Roy, for helping connect with patient advocacy and support groups. They also thank patients and patient advocates Janet Freeman-Daily, Jill Feldman, Ivy Elkins and Tom Carroll for helping them connect with research participants. They also acknowledge the ROSOneder support group, the ALK-Positive Facebook support group and the EGFR Resisters for supporting and promoting this work.

**Contributors** MAA, MZS, FMW, RDN, BHG and MT contributed to literature review and conceptualisation of the work. MAA conducted the interviews. MAA conducted the primary analysis of the data. MZS and MT did peer debriefing and review of analysis with MAA individually and in groups. MAA, MZS, FMW, RDN, BHG and MT all contributed to the writing of the discussion. All authors reviewed and approved the final version of the study.

**Funding** This research is linked to the CanTest Collaborative, which is funded by Cancer Research UK (C8640/A23385), of which FMW is director and RDN and MT are associate directors and coinvestigators. Information, conclusions and opinions expressed in this presentation are of the authors and no endorsement is intended or should be inferred.

**Competing interests** None declared.

**Patient consent for publication** Not required.

**Ethics approval** The study was approved by the University of Washington Institutional Review Board (study number STUDY00005438).

**Provenance and peer review** Not commissioned; externally peer reviewed.

**Data availability statement** Data sharing not applicable as no datasets generated and/or analysed for this study. Data are available upon reasonable request.

**ORCID iDs**
Morhaf Al Achkar http://orcid.org/0000-0002-4160-0550
Monica Zigman Suchsland http://orcid.org/0000-0001-7007-6973
Fiona M Walter http://orcid.org/0000-0002-7191-6476
Richard D Neal http://orcid.org/0000-0002-3544-2744

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
