## [Reviewer comments · BMJ Open]

ARTICLE DETAILS

TITLE (PROVISIONAL)	Experiences along the diagnostic pathway for patients with advanced lung cancer in the United States: A Qualitative Study
AUTHORS	Al Achkar, Morhaf; Zigman Suchsland, Monica; Walter, Fiona; Neal, Richard; Goulart, BH; Thompson, Matthew

VERSION 1 – REVIEW

REVIEWER	Anne Arber University of Surrey, UK
REVIEW RETURNED	20-Nov-2020

GENERAL COMMENTS	This is a well-written article that demonstrates excellent patient and public involvement. It would be interesting to understand how the main author became involved in the study. The focus of the article is on the patient experience of the diagnostic process, which is described in detail following the model of pathways to treatment. However, the interview protocol appears to have questions not directly relevant to the diagnostic process eg How did you find meaning? What would improve your quality of life today? It isn't clear why questions about emotional wellbeing were asked and how this fits with the focus of the article. Figure 2 is not discussed in any detail. More needs to be said about the different pathways and their effect on the patient experience. How many of the patients experienced each of the different pathways? Were there any gender, age or ethnic differences related to the pathway experienced? Where were the delays in diagnosis more likely to occur and how do these factor in to the seven pathways? Some patients report not receiving clear "safety netting" can you identify whether this was unusual, or common. Did it apply to some patients and not to others? When it is stated that the pathways were iterative and circular what does this mean for how Figure 2 is presented? In the discussion there is reference to up-to-date standards of practice and co-ordinating care and communication can this be unpacked and contextualised in terms of the key study findings.
---

REVIEWER	christos Chouaid Service de Pneumologie, CHI Créteil, Inserm U955, UPEC, IMRB, équipe CEpiA, Créteil France
REVIEW RETURNED	22-Nov-2020

GENERAL COMMENTS	Very relevant work congratulations no comment
---

VERSION 1 – AUTHOR RESPONSE

Reviewer: 1

Dr. Anne Arber, University of Surrey

Comments to the Author:

1. This is a well-written article that demonstrates excellent patient and public involvement.

Response: Thank you for the kind remark.

2. It would be interesting to understand how the main author became involved in the study.

Response: we clarified that the main author a stage 4, ALK positive himself. He's also a qualitative researcher by training.

3. The focus of the article is on the patient experience of the diagnostic process, which is described in detail following the model of pathways to treatment. However, the interview protocol appears to have questions not directly relevant to the diagnostic process eg How did you find meaning? What would improve your quality of life today? It isn't clear why questions about emotional wellbeing were asked and how this fits with the focus of the article.

Response: The diagnosis of cancer is part of the experience that the whole project came to understand. Other parts of the projects are already published under the topic, "Unmet needs and opportunities for improving care for patients with advanced lung cancer on targeted therapies (Al Achkar M, Marchand L, Thompson M, Chow LQ, Revere D, Baldwin LM. Unmet needs and opportunities for improving care for patients with advanced lung cancer on targeted therapies: a qualitative study. *BMJ Open*. 2020 Mar 1;10(3):e032639). This paper focuses only on the diagnosis. The questions about meaning are published in the book "Roads to Meaning and Resilience with Cancer". The questions about community support are published in a paper entitled, "Unmet needs and opportunities for improving care for patients with advanced lung cancer on targeted therapies." These questions are not part of the diagnosis experience. It is customary that a qualitative project will collect data that will be organized in multiple manuscripts.

To avoid confusion, we got rid of the detailed protocol since it is already published in our previous work (Al Achkar M, et al. Unmet needs...), and we included the main questions, and follow-up prompts that are relevant to the diagnosis part of the study.

4. Figure 2 is not discussed in any detail. More needs to be said about the different pathways and their effect on the patient experience. How many of the patients experienced each of the different pathways? Were there any gender, age or ethnic differences related to the pathway experienced?

Response: We added more description of the figure. Now it reads, "We noted seven different diagnostic pathways experienced by patients, rather than a single course. These pathways varied primarily by the initial presentation site (primary care, emergency room, etc.) due to the perceived urgency of symptoms."

As a qualitative study with a sample that's not collected to draw inferences about frequencies, we are careful not to suggest an answer to quantitative questions, such as the one posed by the reviewer. This is an exploratory qualitative study, and its goal is to describe the pathways rather than to report on frequencies or conduct quantitative comparisons. We ensure that at least a few have gone on every path. Our study will fall short of reporting the numbers, and we include this as a limitation since it is beyond the scope. However, our team, informed by the findings of this study, is doing research to answer exactly this same question: frequencies of patients who follow this or that pathway. The new project is separate and will be a follow-up study based on different kinds of data.

We added this notion as a limitation, and that reads as, "as a qualitative exploration, our study was not equipped to provide insights about frequencies of occurrences, time indicators, or variations between participants based on their characteristics."

5. Where were the delays in diagnosis more likely to occur and how do these factor in to the seven pathways? Some patients report not receiving clear "safety netting" can you identify whether this was unusual, or common. Did it apply to some patients and not to others?

Response: This is a very important question and will be explored in future studies as it is beyond the

scope of this specific paper. As a qualitative exploration, we don't have the power to draw inferences or compare likelihoods. We included that as a limitation.

The sense in the group is that "safety netting" applied to some but not all. We can generally state that our group is probably more inclusive of people of higher socioeconomic status relative to the broad population of lung cancer patients, but we caution about making comments about the prevalence of this issue. The perception of having no "safety netting" was common but did not apply to all. We clarified that. Our stories are based on patient accounts, and they are the ones who told stories of some providers having contingency and follow up plans. We would, however, caution of putting too much emphasis on the frequency or even calling this as failing due to the nature of the source of our data.

6. When it is stated that the pathways were iterative and circular what does this mean for how Figure 2 is presented?

Response: Figure 2 includes the main points along the pathway. To avoid confusion and added complexity to an already complex figure, we removed the mentioning of iterative and circular. The way the pathways are presented in figure 2 depicts the points along the path rather than the journey of the person who walked the path.

7. In the discussion there is reference to up-to-date standards of practice and co-ordinating care and communication can this be unpacked and contextualised in terms of the key study findings.

Response: This was clarified by adding, "This issue is relevant especially to patients with lung cancer where targeted therapy has changed the disease outcomes in the past few years for patients who have received molecular testing. It's paramount that these new standards of care be available promptly to all patients."

Reviewer: 2

Prof. Christos Chouaid, Centre Hospitalier Intercommunal de Creteil Comments to the Author:

Very relevant work

congratulations

no comment

VERSION 2 – REVIEW

REVIEWER	Anne Arber University of Surrey, UK
REVIEW RETURNED	21-Feb-2021
GENERAL COMMENTS	The authors have responded to all the points raised in the review.